# A Causal Relationship between the New-Type Urbanization and Energy Consumption in China: A Panel VAR Approach

Cheng Chen [1,*], Yajie Gao [1] and Yidong Qin [2]

1    School of Business, Hunan University of Science and Technology, Xiangtan 411201, China; 21301503001@mail.hnust.edu.cn
2    School of Economics, Wuhan University of Technology, Wuhan 430070, China; qyd@whut.edu.cn
*    Correspondence: 20301503001@mail.hnust.edu.cn

**Abstract:** The accelerated urbanization process has been considered to be the root cause of increasingly severe energy consumption growth in China. However, energy is still an essential factor for the urbanization process, so arbitrarily mitigating energy use currently will unquestionably slow down the urbanization process. The principal contribution of this paper is to comprehensively analyze the dynamic interaction mechanism between the new-type urbanization and energy consumption, and further put forward a new idea of comparing the benefit of an increase in the level of new-type urbanization resulting from energy consumption and the negative externality of environmental damage related with energy consumption. This paper conducts an empirical study on the causal relationship between new-type urbanization and energy consumption using Chinese provincial administrative units from 1999 to 2020. And we find that new-type urbanization leads to energy consumption negatively and energy consumption leads to new-type urbanization positively for provinces in the eastern region. There is only a one-way effect of energy consumption on new-type urbanization for provinces in the central and northeastern regions, and there is negative feedback causality for provinces in the western region. Additionally, the benefit of an increase in the level of new-type urbanization resulting from energy consumption is larger than the negative externality of environmental damage related to energy consumption for provinces in the eastern, central, and northeastern regions, yet it is totally opposite for provinces in the western region. Finally, we propose some fruitful policy recommendations to construct new-type urbanization under the background of clear reduction targets for energy consumption in China.

**Keywords:** new-type urbanization; energy consumption; environmental Kuznets curve; PVAR approach



## 1. Introduction

When the "Reform and Opening-up" policy was put into practice in the year of 1978, China's urbanization experienced a steady upward trend [1,2]. According to the statistics from the National Bureau of Statistics (NBS), China's urbanization rate increased from 17.9% in 1978 to 65.2% in 2022. Nevertheless, much of China remains to be urbanized, especially the inland regions. As China has a larger population, millions of rural residents will move into urban areas every year during to-be urbanization process. Although China's urbanization greatly raises people's living standards, it has also been deemed to be the root cause of increasingly severe energy consumption growth; for example, the heavy use of fossil energy, which undoubtedly brings about a series of environmental pollution [3]. The total energy consumption in China even made up 23.6% of the global total in the year 2018 [4]. Specifically, urban areas solely account for 75.15% of total energy consumption in China [5]. As a responsible country, China has been determined to set clear reduction targets for energy consumption. Because the urbanization process in China is still accelerating at present, energy is still an essential factor for the urbanization process. Arbitrarily mitigating

energy use currently will unquestionably slow down the urbanization process. Therefore, there are two kinds of opposite effects simultaneously, which are the benefit of an increase in the level of urbanization resulting from energy consumption and the negative externality of environmental damage related to energy consumption respectively. Could the advantages sufficiently offset the disadvantages? Investigating these two kinds of opposite effects has vital theoretical and practical significance for promoting the urbanization process under the background of clear reduction targets for energy consumption in China. If the advantages of energy consumption outweigh its disadvantages, it pays to promote the urbanization process by increasing energy consumption. However, if energy consumption does not promote or even adversely affect the urbanization process, an energy conservation policy should be adopted to offset the negative externality of environmental damage related to energy consumption.

Previous studies have extensively investigated this subject, yet the majority emphasized the one-way effect of urbanization on energy consumption in the beginning. Many scholars found that the urbanization process leads to energy consumption growth [6–9]. In contrast, some scholars argued that urbanization may lead to energy consumption negatively [10–16]. Based on Northam [17], the third strand of research further examined the nonlinear relationship between the two due to the mixed findings. Most scholars confirmed that the nexus of the two is indeed nonlinear [2,18–20]. Another branch of the literature used city size as a proxy variable for urbanization rate, which indicated that there also was a nonlinear relationship between city size and energy consumption [21,22]. There are two kinds of explanations for the nonlinear relationship between the two. On the one hand, the essence of urbanization can be ascribed to the agglomeration effect, scale effect, and spatial spillover effect, which are conducive to reducing the energy consumption of residents or increasing energy efficiency [5,14,23–25]. On the other hand, the urbanization process reduces energy consumption through industrial structure upgrading and technical innovation [19,26–28].

The level of urbanization is mainly measured by the single index method in the above studies. These simple indicators can only mirror the level of population-oriented urbanization rather than the improvement of production and lifestyle. Especially after China issued the "National New-type Urbanization Plan" in 2014, human-centered urbanization has been put into practice. Therefore, investigating the relationship between new-type urbanization and energy consumption has much more practical significance currently. Recently, a few scholars used the comprehensive index method to measure its connotation and further examine its effect on energy consumption. Liu et al. [14] used a spatial econometric model for China's regions on this subject and found that new-type urbanization leads energy consumption negatively, yet its effect on adjacent areas or the spatial spillover effect is positive. Lin and Zhu [4] examined the effect of new-type urbanization on energy saving and its transmission channels based on Chinese cities and found that it can bring about an energy-saving effect. Yu [29] examined the ecological effect of new-type urbanization and found that China's new-type urbanization can improve energy efficiency. Feng et al. [30] examined the effect of new-type urbanization on energy efficiency based on Chinese cities and found that it has a double-threshold effect. Shao and Wang [31] examined the effect of new-type urbanization on green total factor energy efficiency and found that it has a heterogeneous effect for different cities. Not surprisingly, the relationship between the two is much more complicated compared to traditional urbanization.

Apart from that, as an essential factor of economic development, energy consumption is also conducive to promoting the level of urbanization. Ghosh and Kanjilal [32] investigated the cointegration relationship between the two for India and found that there is causality running from energy consumption to urbanization. Wu et al. [2] estimated the direct effects of various energy consumption patterns in China and found that energy consumption leads to urbanization positively, and the positive effect is dependent on energy consumption intensity, energy consumption scale, and energy consumption structure. Xu and Wang [33] examined the threshold effect of energy consumption on new-type

urbanization in China and found that there was a significant threshold effect. Some scholars further explored how energy consumption affected the urbanization process. These studies found that the carbon emission reduction effect [34], agglomeration economy effect and economies of scale [32,35,36], and the industrial structure effect [36–38] are primary transmission channels through which energy consumption affects urbanization.

The above studies in this field indicate that there should be a bi-directional causality between the two. So far, a wealth of studies primarily investigated uni-directional causality on this subject, yet the bi-directional causality between the two is still scarce. Comparatively, a large body of studies proved that economic growth and energy consumption present a bi-directional causal relationship [39–44]. As urbanization is widely considered a symbol of economic development [37–39], an abundance of support can be indirectly found for the bi-directional causality relationship on our subject. To the best of our knowledge, only Tang et al. [45] explored the two-way correlation mechanism between new-type urbanization and clean energy consumption based on Chinese provincial data and found that there is a significant two-way promoting effect between the two. To sum up, the extant literature in this field actually denotes that energy consumption probably affects urbanization by means of its effects on economic development.

Despite the existing studies in this field having explored extensively the relationship between the two, there are still a few drawbacks on this subject. Firstly, the bulk of empirical studies simply examined how urbanization affects energy consumption, or whether energy consumption promoted the level of urbanization, and these empirical results did not compare the benefit of an increase in the level of urbanization resulting from energy consumption and the negative externality of environmental damage related to energy consumption, so they cannot provide corresponding policy implications for promoting the urbanization process. Secondly, there should be a bi-directional causality between the two, and the existing studies mainly examined the one-way effect of urbanization on energy consumption. Thirdly, the empirical studies on this subject are generally conducted based on the linear relationship hypothesis, and the estimated results are always inconsistent. The nonlinear relationship hypothesis may be more realistic, especially for the effect of new-type urbanization, which remains to be further examined empirically. In this regard, our study makes the following contributions. Firstly, this paper comprehensively analyzes the dynamic interaction mechanism between new-type urbanization and energy consumption, aiming to reveal the bi-directional causality between the two and extend the depth and breadth of this subject. Secondly, this paper put forward a new idea of comparing the benefit of an increase in the level of new-type urbanization resulting from energy consumption and the negative externality of environmental damage related to energy consumption, which can provide targeted policy recommendations. Thirdly, this paper creatively explains the estimated results with the concept of the well-known Environmental Kuznets Curve (EKC) hypothesis based on some energy-related data, which not only adds new empirical evidence for the EKC relation but also provides a robustness test for our regression results given that new-type urbanization and energy consumption factually present the two-way causal relationship.

The remainder of this article is arranged as follows. The following section introduces the measurement methods and analysis of the measurement results; Section 3 outlines the econometric specification and presents empirical results; and Section 4 concludes.

## 2. Measurement Methods and Analysis of Measurement Results

### 2.1. Measurement Methods

The most critical things are methods for calculating the level of new-type urbanization and energy consumption in this paper. According to Ma et al. [46], the amount of energy consumption per capital is applied to measure the level of energy consumption. Similar to the existing studies [4,47], the composite index method is used to fully measure the level of new-type urbanization. The index system is made up of two levels, which include a total of 19 computable indices, as is shown in Table 1. To overcome some shortcomings

of the subjective weighting method, the entropy method is employed to calculate the constructed comprehensive urbanization index. The "+" and "−" of Index Attributes in Table 1 signify the influence of 19 computable indices on the comprehensive index, and "+" indicates an increase in the metric of indices would promote the comprehensive index, and "−" indicates an increase in the metric of indices would decrease the comprehensive index.

**Table 1.** China's comprehensive urbanization index system. Reprinted from [48]. Copyright 5584080645167 (2023) with permission from Elsevier.

| Index I | Index II | Index Attribute |
|---|---|---|
| Population | Urban population density | + |
| | Full-time equivalent of R&D personnel | + |
| | Urban population ratio | + |
| | Number of college degrees or above per ten thousand people | + |
| | Proportion of employed persons in the tertiary industry | + |
| Economy | Gross domestic products per capita | + |
| | Consumption proportion of urban to rural residents | − |
| | Disposable income of urban household per capita | + |
| | Tertiary industry as a percentage of regional GDP | + |
| Living environment | Urban wastewater treatment ability per day | − |
| | Green covered area as a percentage of completed area | + |
| | Greenery area of per capital park | + |
| | Area under a cleaning program per square kilometer of built-up area | + |
| Living conditions | Urban gas access rate | + |
| | Urban water access rate | + |
| | Number of public toilets per ten thousand people | + |
| | Number of public transportation vehicles per ten thousand people | + |
| | Urban per capita area of paved roads | + |
| | Number of patent grants per ten thousand people | + |

The sample data of our empirical research spans 22 years, from 1999 to 2020, and includes all provincial administrative units in China. The original data for energy consumption are obtained from the China Energy Statistical Yearbook and the China Statistical Yearbook. All relevant data for calculating the comprehensive urbanization index are obtained from the China Statistical Yearbook, China Energy Statistical Yearbook, China City Statistical Yearbook, China Statistical Yearbook on Science and Technology, each provincial statistical yearbook, and so on. Considering the data unavailability, Tibet, Hong Kong, Macao, and Taiwan are deleted from the sample in our empirical study. Consequently, the research sample finally consists of 30 provincial administrative units. And these 30 provincial administrative units can be classified into four categories according to the National Bureau of Statistics: east, northeast, central, and west. So as to cancel the impact of the price level in different years, all data related to nominal GDP are revised to a constant price based on the 1999 price index in the process of computation.

*2.2. Analysis of Measurement Results*

According to the specific calculation method introduced above, the provincial level of energy consumption in China from 1999 to 2020 is calculated firstly. To visually describe the characteristics of the temporal–spatial evolution in the provincial level of energy consumption, the measurement results are reported in the form of topographic maps. Figure 1 shows the concrete results. Due to space constraints, the results from 1999, 2007, 2013, and 2020 are only displayed. It can be clearly seen in Figure 1 that the dark blue areas represent the highest level of energy consumption in 30 provincial administrative units. Obviously, the

number of dark blue regions increased from zero in 1999 to ten in 2020. For the convenience of discussions, all provincial administrative units are called "province". Therefore, we can conclude from these results that the provincial level of energy consumption in China is on the rise over the study period [43,49]. And in 2020, the provincial level of energy consumption demonstrated visible spatial differentiation. Among the provinces with the highest level of energy consumption, four provinces lie in the western region: Xinjiang, Qinghai, Gansu, and Inner Mongolia. Four provinces, Beijing, Shanghai, Hebei, and Jiangsu, are in the eastern region. Only one province, Shaanxi, lies in the central region, and only one province, Liaoning, lies in the northeastern region. Generally, the western region has the highest level of energy consumption among the four types of regions, followed by the eastern region. And the rest of the two regions have comparatively lower levels of energy consumption. The provincial level of energy consumption from the eastern to the central, northeastern, and western regions is similar to the U-shaped curve [49]. The spatial characteristics of the provincial level of energy consumption imply preliminarily that it is necessary to give thought to regional heterogeneity on this subject.

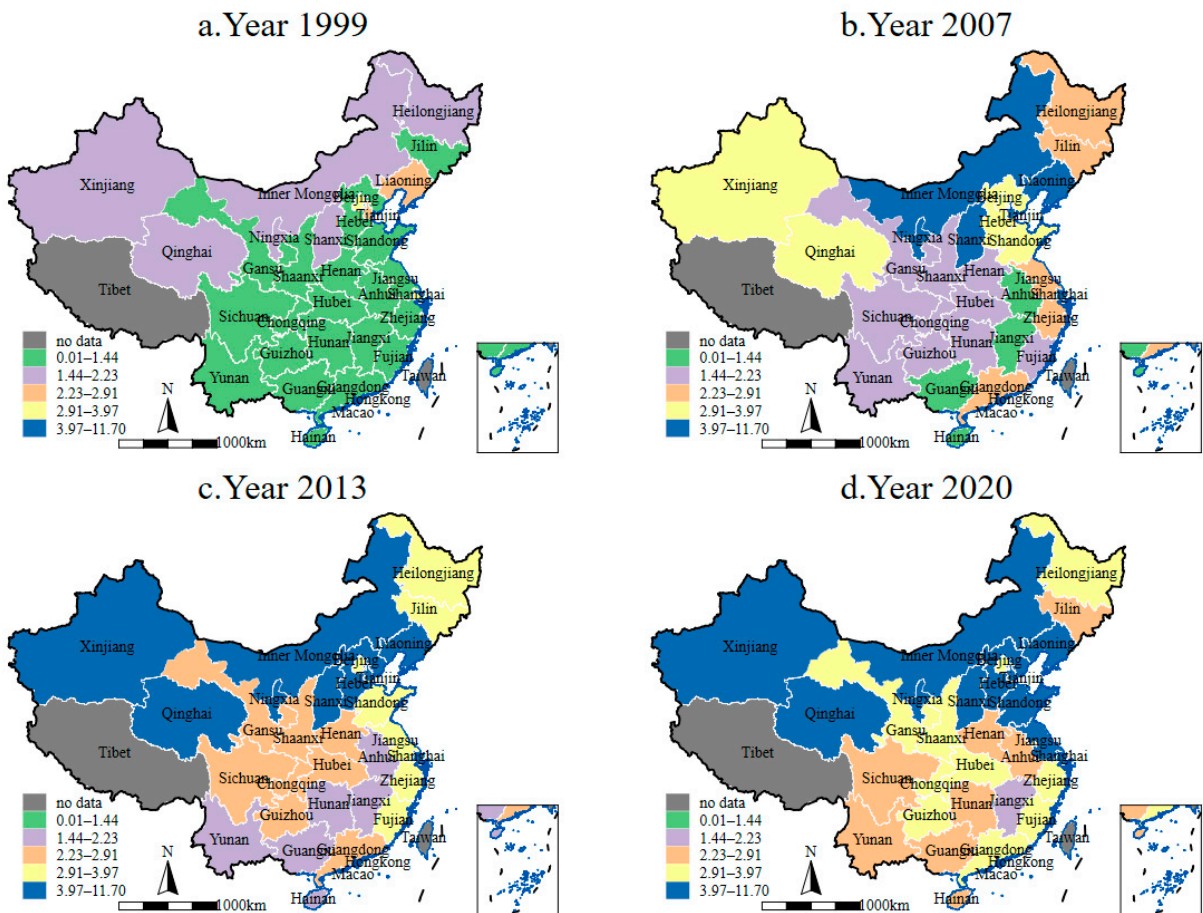

**Figure 1.** Chinese provincial level of energy consumption for some years (million tons).

Then, the comprehensive urbanization index is calculated by the widely used entropy method. To visually describe the characteristics of the temporal–spatial evolution of this index, the measurement results are reported similarly in the same way. Figure 2 shows the concrete results. Due to space constraints, the results from 1999, 2007, 2013, and 2020 are only displayed. It can be clearly seen in Figure 2 that the dark blue areas represent the highest level of new-type urbanization in 30 provincial administrative units. Obviously, the number of dark blue regions is zero in 1999, yet nearly all provinces in the eastern and central regions belonged to the highest group in 2020. Therefore, we can conclude from these results that the provincial level of new-type urbanization has also

increased greatly over the study period, especially after the year 2013 [2,4,49]. On the one hand, the main reason for these time series characteristics can be probably ascribed to the "National New-type Urbanization Plan", which makes the economy, society, and ecology balanced in the process of urbanization. On the other hand, the increase in the level of energy consumption may also be conducive to speeding up the process of urbanization. And in 2020, the provincial level of new-type urbanization has also displayed significant spatial differentiation. Specifically, the majority of provinces have achieved the highest status in 2020. And only a few provinces have not reached the highest level, these are Xinjiang, Jilin, Shanxi, Gansu, Guizhou, Qinghai, Yunnan, and Guangxi. Additionally, these provinces mainly lie in the western and central regions. The average comprehensive urbanization index of the provinces in the eastern, central, northeastern, and western regions is calculated to be 0.299, 0.169, 0.188, and 0.159, respectively, which verifies that the provincial level of new-type urbanization looks like the inverted S-shaped curve, which is slightly different with the provincial level of energy consumption [46,47]. Therefore, the spatial characteristics of the provincial level of new-type urbanization further imply that it is necessary to give thought to regional heterogeneity on this subject.

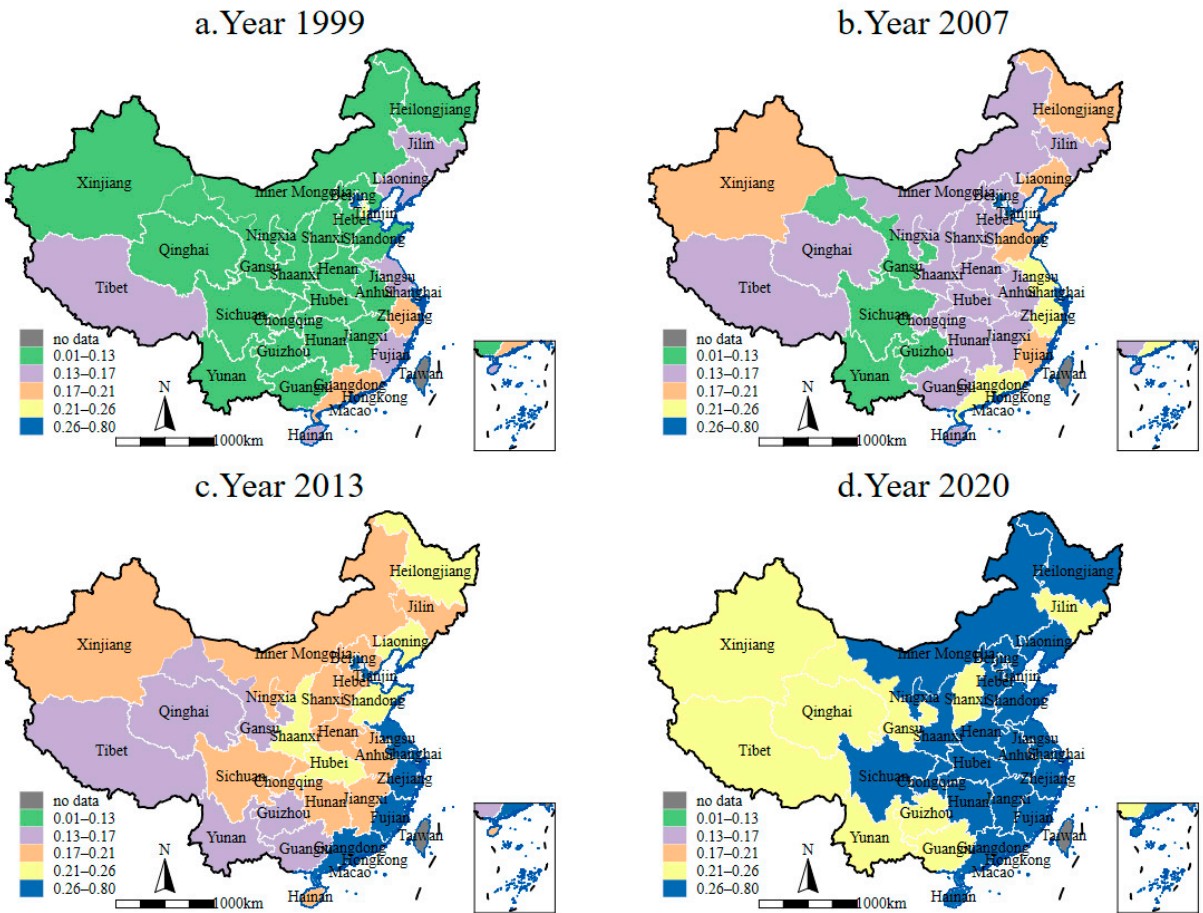

**Figure 2.** Chinese provincial level of new-type urbanization for some years.

To display the tendency intuitively, the growth rate of the above measured variables is depicted with a scatter diagram in Figure 3 for the four groups, eastern (Figure 3a), central (Figure 3b), northeastern (Figure 3c), and western regions (Figure 3d), respectively. In Figure 3, the relationship between comprehensive urbanization index growth and energy consumption growth is significantly different in the four groups. Specifically, there seems to be a strong relationship between the northeast and western regions, yet no such relationship emerges in the eastern and central regions. These results further prove that we cannot neglect the "one size for all" homogeneity issue among provinces when investigating

the causal relationship between the two, and it is reasonable to classify the data into four groups: east, northeast, central, and west. Secondly, the weak relationship between the two in the eastern and central regions denotes there may be a nonlinear relationship between comprehensive urbanization index growth and energy consumption growth, and the traditional linear model is not suitable for this subject. Nevertheless, it should be noted that these kinds of scatter charts roughly reflect the possible correlation relationship between the two, and the exact relationship between the two remains to be confirmed rigorously by employing reasonable econometric models.

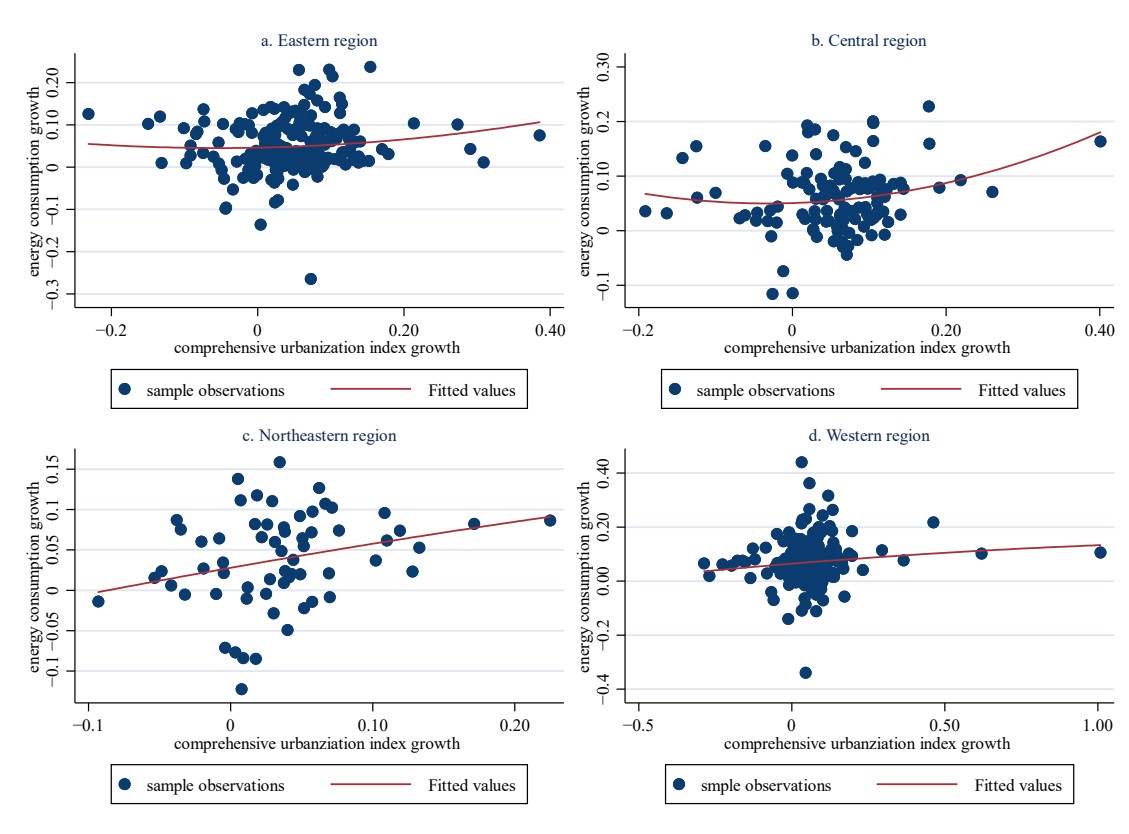

**Figure 3.** Comprehensive urbanization index growth vs. energy consumption growth.

## 3. PVAR Model Regression Results and Analysis

### 3.1. Model Specification

So as to fully reveal the causal on this subject, the panel vector autoregression (PVAR) model is employed to conduct empirical research. Compared with the widely used vector autoregression (VAR) model, the PVAR model has the advantage of dealing with long-term panel data and endogenous causality in the traditional linear regression model. The corresponding PVAR model is constructed as follows:

$$Y_{it} = \Gamma_0 + \sum_{p=1}^{n} \Gamma_p Y_{it\text{-}p} + \delta_i + f_t + \varepsilon_{it}$$

where the subscripts *i* and *t* denote province and year, respectively; $Y_{it}$ is a multi-dimensional variable, which is *NU* and *EC*, representing the level of new-type urbanization and energy consumption, respectively, measured by the methods introduced hereinabove; $Y_{it\text{-}p}$ is a *p*-period lag term of $Y_{it}$; and $\delta_i$ and $f_t$ indicate individual province fixed effects and year fixed effects, respectively. $\varepsilon_{it}$ is termed the random disturbance.

As the PVAR model includes the lag term of dependent variables and individual province fixed effects, it can be considered a typical dynamic panel data model. Thus, the

OLS estimation will be biased and inconsistent. Therefore, the system GMM proposed by Blundell and Bond [49] is used to estimate the PVAR model. The PVAR model is commonly conducted by the following steps [50,51]. Firstly, the stationarity of the panel data and the causal relationship between *NU* and *EC* have to be tested. Secondly, the optimal lag order needs to be selected, and the PVAR model can be estimated logically. Thirdly, the impulse response graph will be displayed based on the PVAR model regression. The last step is the variance decomposition.

### 3.2. PVAR Model Regression Analysis

#### 3.2.1. Stationarity Tests

First of all, it is necessary to inspect the stationarity of time series variables. The IPS, Fisher-ADF, and Fisher-PP are comprehensively used to implement unit root tests, and the results are reported in Table 2. In Table 2, the *p*-values of all tests for the first-difference series of *NU* and *EC* are all 0.000, which rejects the null hypothesis at the 1% significance level. Hence, we can conclude that the first-difference series of *NU* and *EC* are stationary. In other words, these original data are integrated processes of order one.

**Table 2.** Unit root test results.

| Variables | IPS | | Fisher-ADF | | Fisher-PP | |
|---|---|---|---|---|---|---|
| | Statistics | *p*-Value | Statistics | *p*-Value | Statistics | *p*-Value |
| *NU* | −3.272 | 0.001 | 107.964 | 0.000 | 39.533 | 0.988 |
| $NU_{-1}$ | −13.091 | 0.000 | 290.177 | 0.000 | 615.121 | 0.000 |
| *EC* | −1.569 | 0.363 | 141.268 | 0.000 | 45.468 | 0.943 |
| $EC_{-1}$ | −10.008 | 0.000 | 219.605 | 0.000 | 290.094 | 0.000 |

Notes: The null hypothesis is that the time series variables have a unit root process.

#### 3.2.2. Benchmark Regression Results

Before estimating the PVAR model, an optimal lag *p*-period of time series variables remains to be explored. Based on the standard procedure in empirical studies, the AIC, BIC, and HQIC are used to select the optimal lag order. The selection of two lag periods is reasonable. Therefore, 2 is the optimal lag order in this paper. The estimation results of the PVAR model are reported in Table 3. The *EC* equation reflects the effects of *EC* and *UN* on *EC*, and the *UN* equation reflects the effects of *UN* and *EC* on *UN*. The estimated results for all provinces as a whole are reported in the top line of Table 3. As shown in Table 3, in the *EC* equation, the first-period lag of *EC* has a significant positive effect on *EC*, and the second-period lag of *EC* has a significant negative effect on *EC*, which indicates that energy consumption presents the characteristics of path-dependent inertia in the short run, yet it tends to converge in the long run [4,47]. The first-period lag of *NU* has a significant negative effect on *EC* and the second-period lag of *NU* has a significant positive effect on *EC*, which indicates that new-type urbanization leads to energy consumption negatively in the short run [10–16], yet the new-type urbanization leads to energy consumption positively in the long run [6–9]. In the *NU* equation, the first-period lag of *NU* has a significant positive effect on *NU*, and the second-period lag of *NU* does not have any significant effect on *NU*, which indicates that new-type urbanization also presents the characteristics of path-dependent inertia in the short run. The first-period lag of *EC* has a significant positive effect on *NU* [33], and the second-period lag of *EC* has a significant negative effect on *NU* [45], which indicates that an increase in energy consumption brings about a further increase in the level of new-type urbanization in the short run, yet it is detrimental to new-type urbanization in the long run.

**Table 3.** Estimation results of the PVAR model.

| Sample | Type | Variable | Coefficient | Variable | Coefficient |
|---|---|---|---|---|---|
| Countrywide | *EC* equation | $EC_{-1}$ | 1.553 ***<br>(14.04) | $NU_{-1}$ | −1.279 **<br>(2.00) |
| | | $EC_{-2}$ | −0.513 ***<br>(−8.32) | $NU_{-2}$ | 1.138 *<br>(1.81) |
| | *NU* equation | $NU_{-1}$ | 1.342 ***<br>(4.84) | $EC_{-1}$ | 0.021 **<br>(2.37) |
| | | $NU_{-2}$ | −0.257<br>(−0.84) | $EC_{-2}$ | −0.022 ***<br>(−2.58) |
| Eastern | *EC* equation | $EC_{-1}$ | 1.492 ***<br>(15.46) | $NU_{-1}$ | −0.879 *<br>(−1.80) |
| | | $EC_{-2}$ | −0.465 ***<br>(−5.45) | $NU_{-2}$ | 0.638<br>(1.18) |
| | *NU* equation | $NU_{-1}$ | 1.083 ***<br>(5.09) | $EC_{-1}$ | 0.042 ***<br>(4.42) |
| | | $NU_{-2}$ | 0.008<br>(0.04) | $EC_{-2}$ | −0.035 ***<br>(−4.80) |
| Central | *EC* equation | $EC_{-1}$ | 1.451 ***<br>(11.31) | $NU_{-1}$ | −1.188<br>(−1.40) |
| | | $EC_{-2}$ | −0.495 ***<br>(−4.21) | $NU_{-2}$ | 1.321 *<br>(1.69) |
| | *NU* equation | $NU_{-1}$ | 1.507 ***<br>(7.37) | $EC_{-1}$ | 0.031 **<br>(2.35) |
| | | $NU_{-2}$ | −0.448 **<br>(−2.02) | $EC_{-2}$ | −0.027 **<br>(−2.02) |
| Northeastern | *EC* equation | $EC_{-1}$ | 1.689 ***<br>(10.52) | $NU_{-1}$ | −10.142<br>(−1.35) |
| | | $EC_{-2}$ | −0.408 ***<br>(−3.43) | $NU_{-2}$ | 5.158 *<br>(1.87) |
| | *NU* equation | $NU_{-1}$ | 0.891 ***<br>(3.59) | $EC_{-1}$ | 0.029 ***<br>(2.81) |
| | | $NU_{-2}$ | 0.189<br>(0.94) | $EC_{-2}$ | −0.026 ***<br>(−2.93) |
| Western | *EC* equation | $EC_{-1}$ | 1.508 ***<br>(18.47) | $NU_{-1}$ | −3.508 ***<br>(−3.05) |
| | | $EC_{-2}$ | −0.415 ***<br>(−5.21) | $NU_{-2}$ | 1.462 *<br>(1.95) |
| | *NU* equation | $NU_{-1}$ | 1.339 ***<br>(4.57) | $EC_{-1}$ | 0.010<br>(2.37) |
| | | $NU_{-2}$ | −0.284<br>(−0.97) | $EC_{-2}$ | −0.008 *<br>(−1.66) |

Notes: ***, **, and * show significant levels at 1%, 5%, and 10%, respectively. The value given in parentheses is *t* statistics. The subscripts $_{-1}$ and $_{-2}$ represent the first-period lag and the second-period lag, respectively.

The estimated results for all provinces as a whole neglect regional heterogeneity. To explore this issue, groups of provinces are classified into four types of regions according to NBS. The estimated results for provinces in the eastern region are reported in the top second line of Table 3. As shown in Table 3, in the *EC* equation, the first-period lag of *EC* and the second-period lag of *EC* have a similar effect on *EC* compared to the whole country. The first-period lag of *NU* has a significant negative effect on *EC*, and the second-period lag of *NU* does not have any effect on *EC*, which indicates that new-type urbanization leads to energy consumption negatively in the short run [4,30], yet this inhibitory effect gradually disappears over time. In the *NU* equation, both *NU* and *EC* have a similar effect on *UN* compared to the whole country.

The estimated results for provinces in the central region are reported in the top third line of Table 3. As shown in Table 3, in the *EC* equation, the first-period lag of *EC* and the

second-period lag of *EC* have a similar effect on *EC* compared to the whole country. Both the first-period lag of *NU* and the second-period lag of *NU* do not have any significant effect on *EC*, which indicates that an increase in the level of new-type urbanization does not bring about energy consumption [52]. In the *NU* equation, the first-period lag of *NU* has a significant positive effect on *NU,* and the second-period lag of *NU* has a significant negative effect on *NU*. The first-period lag of *EC* and the second-period lag of *EC* have a similar effect on *UN* compared to the whole country.

The estimated results for provinces in the northeastern region are reported in the fourth line of Table 3. As shown in Table 3, in the *EC* equation, the first-period lag of *EC* and the second-period lag of *EC* have a similar effect on *EC* compared to the whole country. Both the first-period lag of *NU* and the second-period lag of *NU* do not have any significant effect on *EC*, which also indicates that an increase in the level of new-type urbanization does not bring about energy consumption [52]. In the *NU* equation, both *NU* and *EC* have a similar effect on *UN* compared to the whole country.

The estimated results for provinces in the western region are reported in the fifth line of Table 3. As shown in Table 3, in the *EC* equation, the first-period lag of *EC* and the second-period lag of *EC* have a similar effect on *EC* compared to the whole country. The first-period lag of *NU* and the second-period lag of *NU* have a similar effect on *EC* as the eastern region, which indicates that new-type urbanization leads to energy consumption negatively in the short run [4,30], yet this inhibitory effect gradually disappears over time. In the *NU* equation, the first-period lag of *NU* and the second-period lag of *NU* have a similar effect on *NU* compared to the whole country. The first-period lag of *EC* does not have any effect on *UN,* and the second-period lag of *EC* has a significantly negative effect on *UN*, which indicates that an increase in energy consumption is detrimental to new-type urbanization.

### 3.2.3. Discussions with Concept of EKC

According to existing studies, energy consumption may lead to economic development and environmental pollution simultaneously [53–55]. Since new-type urbanization is widely considered a symbol of economic development [38], the critical issue is whether energy consumption can bring about larger benefits with respect to its cost. This basic benefit–cost tradeoff can be inferred from the causal relationship between the two. When energy consumption is conducive to promoting the level of new-type urbanization, it may indicate that the benefit of an increase in the level of new-type urbanization resulting from energy consumption is larger than the negative externality of environmental damage related to energy consumption. On the contrary, if the new-type urbanization leads to energy consumption positively, it may indicate that the advantage of energy consumption exceeds its disadvantage. From the estimated results for all provinces as a whole, we can conclude that the advantage of energy consumption is larger than its disadvantage in the short run, yet the relationship is opposite over time. From the estimated results for provinces in the eastern region, we can conclude that the advantage of energy consumption is always larger than its disadvantage over time. This again proves that those provinces in the eastern region may have started to cope with the possible environmental damage related to energy consumption. From the estimated results for provinces in the central, northeastern, and western regions, the advantages of energy consumption and its disadvantages are similar to the whole country, where the negative externality of environmental damage exceeds its benefit over time.

The PVAR model regression results can also be interpreted with the concept of the EKC, which assumes that economic growth and environmental pollution present an "inverted U" relationship [56]. Initially, as the level of new-type urbanization is relatively low, there are not too many industrial activities that lead to environmental pollution. Therefore, an increase in the level of new-type urbanization is conducive to reducing energy consumption for all samples in the short run. As the pace of new-type urbanization accelerates, there will be more and more high-pollution industries, and environmental pollution

may gradually increase. As the estimation results indicated, an increase in the level of new-type urbanization may enhance energy consumption in the long run for the central, northeast, and western regions. Even so, for provinces in the eastern region, new-type urbanization is not conducive to increasing energy consumption over time. As the economy improves, these provinces may start to focus on the possible environmental damage related to energy consumption and attempt to take some remedial actions [57]. Generally speaking, as long as new-type urbanization reaches a high level as the eastern region, more resources may be dedicated to environmental protection. Consequently, an increase in the level of new-type urbanization will be conducive to reducing environmental pollution. Our main target is not to explore the EKC, yet the implications of the estimated results are compatible with the EKC prediction. To sum up, as the pace of new-type urbanization accelerates, a negative externality, such as environmental pollution related to energy consumption, gradually increases. Once a province reaches a high level of new-type urbanization, it may conversely reduce the negative externality related to energy consumption, as indicated by the EKC assumption.

### 3.3. Impulse Response Analysis

So as to further investigate the dynamic relationship between the variables in the PVAR model, the impulse response function is computed. Specifically, we set up Monte Carlo simulations for four types of subsamples and finally obtained a 4 × 4 impulse response graph, and the corresponding results are displayed in Figure 4 for the different groups: eastern (Figure 4a), central (Figure 4b), northeastern (Figure 4c), and western regions (Figure 4d), respectively.

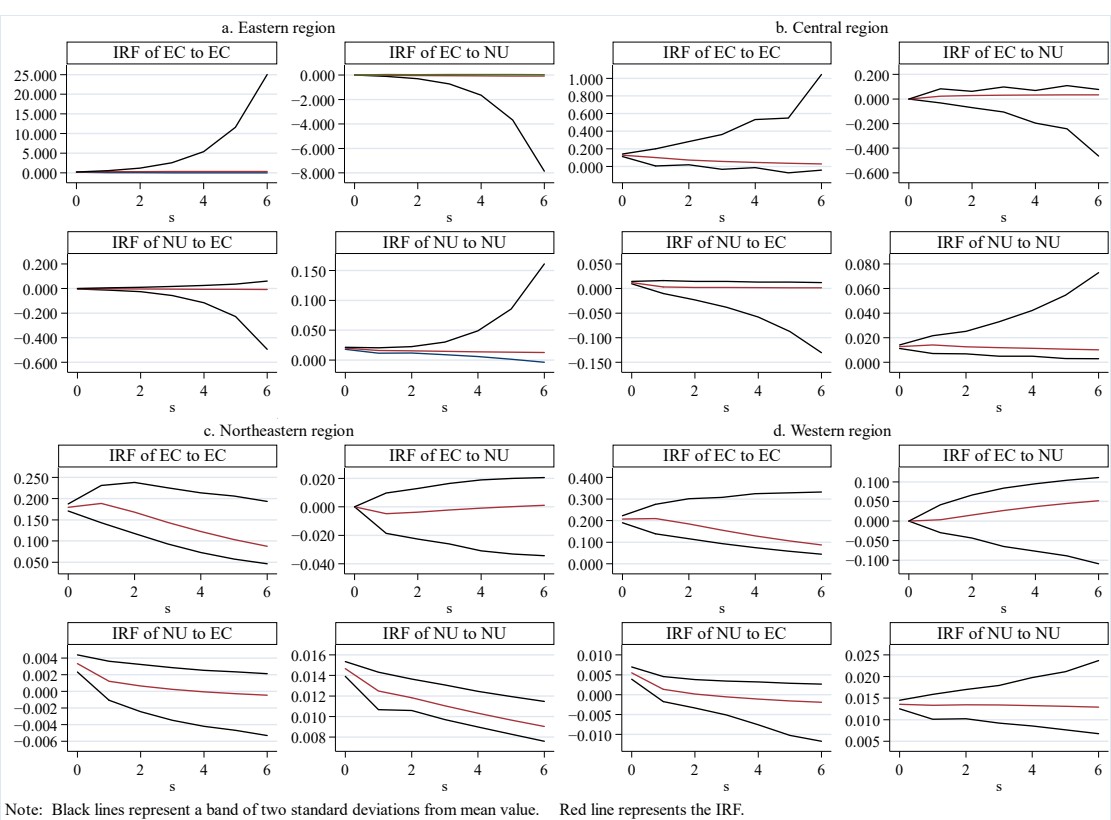

**Figure 4.** Variable impulse responses.

For provinces in the eastern regions, only *NU* shows a positive response to a standard deviation in its own unit. These results further verify that provinces in the eastern region may have been committed to environmental protection, hence there is no dynamic relationship. For provinces in the central region, the impulse response of *EC* to *NU* is positive,

which is consistent with the PVAR model regression analysis. The implication is that these provinces should pay much attention to the high-pollution industries while the pace of new-type urbanization accelerates. The impulse response of *NU* to *EC* is positive in the current period, and then it decreases in the first period and tends to be weak after the second period, which is also consistent with the PVAR model regression analysis. The implication is that the negative externality of environmental damage related to energy consumption may offset the benefit of an increase in the level of new-type urbanization from energy consumption over time. While energy consumption is not conducive to promoting the level of new-type urbanization, these provinces should adopt some conservation policies. For provinces in the northeastern region, the impulse response of *EC* to *NU* is negative, which quickly reaches the highest value and then converges to zero. The implication is that the negative externality of environmental damage related to energy consumption may gradually increase over time. The impulse response of *NU* to *EC* is positive in the current period, and then it decreases in the first period and tends to be weak after the fourth period, which is the same as the provinces in the central region. For the provinces in the western region, the impulse response of *EC* to *NU* and the impulse response of *NU* to *EC* are virtually the same as the provinces in the central region, hence it is not necessary to interpret these results. In contrast to other regions, it is noteworthy that the impulse response of EC to NU does not diminish over time. This further denotes that the urbanization process in the western region relies predominantly on high energy consumption and highly pollutant-intensive inputs, and the legacy of conventional economic development approaches remains an arduous obstacle to overcome.

### 3.4. Variance Decomposition

In order to compare the relative contribution degree of unit standard deviation in *EC* and *NU*, we further adopt the variance decomposition. The variance decomposition results of period 5, period 10, and period 20 are displayed in Table 3 for different groups: (a) eastern, (b) central, (c) northeastern, and (d) western regions. As shown in Table 4, for provinces in the eastern, central, and western regions, the variance decomposition of *EC* is dominated by its own shock, and *NU* has a small effect. However, for provinces in the northeastern region, the variance decomposition of *EC* is dominated by the effect of *NU*, and its own shock has a small effect. For provinces in the eastern region, the variance decomposition of *NU* to its own shock is almost equal to the effect of *EC*. However, for provinces in central, northeastern, and western regions, the variance decomposition of *NU* is dominated by its own shock, and *EC* has a small effect. Combined with the variance decomposition of both variables, the effect of *EC* on *NU* is much larger than the effect of *NU* on *EC* for provinces in the eastern region, yet it is the opposite for provinces in central, northeastern, and western regions. These variance decomposition results also further confirm that energy consumption may bring about greater advantages relative to its disadvantage for provinces in the eastern region, yet it is imperative to cope with the possible cost of environmental pollution related to energy consumption for the provinces in central, northeastern, and western regions.

**Table 4.** Variance decomposition.

| Response Variable | s | Eastern | | Central | | Northeastern | | Western | |
|:---:|:---:|:---:|:---:|:---:|:---:|:---:|:---:|:---:|:---:|
| | | *EC* | *NU* | *EC* | *NU* | *EC* | *NU* | *EC* | *NU* |
| *EC* | 5 | 0.979 | 0.021 | 0.916 | 0.084 | 0.443 | 0.557 | 0.986 | 0.014 |
| *NU* | 5 | 0.057 | 0.943 | 0.177 | 0.823 | 0.233 | 0.767 | 0.036 | 0.964 |
| *EC* | 10 | 0.950 | 0.050 | 0.818 | 0.182 | 0.255 | 0.745 | 0.913 | 0.087 |
| *NU* | 10 | 0.194 | 0.806 | 0.124 | 0.876 | 0.118 | 0.882 | 0.032 | 0.968 |
| *EC* | 15 | 0.924 | 0.076 | 0.756 | 0.244 | 0.160 | 0.840 | 0.820 | 0.180 |
| *NU* | 15 | 0.360 | 0.640 | 0.106 | 0.894 | 0.069 | 0.931 | 0.040 | 0.960 |
| *EC* | 20 | 0.903 | 0.097 | 0.724 | 0.276 | 0.103 | 0.897 | 0.744 | 0.256 |
| *NU* | 20 | 0.507 | 0.493 | 0.099 | 0.901 | 0.043 | 0.957 | 0.047 | 0.953 |

So as to visually display the relative contribution degree of unit standard deviation in *EC* and *NU*, we further draw the variance decomposition results of *EC* and *NU* in the form of a coordinate axis. Figure 5 shows the variance decomposition of *EC* to *NU* for different groups: variance decomposition of *EC* to *NU* (Figure 5a) and variance decomposition of *NU* to *EC* (Figure 5b), respectively. As shown in Figure 5a, *NU* has a remarkable influence on *EC* for the provinces in the northeastern region, and the contribution rate is the largest among the four groups. Additionally, *NU* has an increasing contribution rate to unit standard deviation in *EC* for all groups. As explained above, these results imply that it is imperative to cope with the possible cost of environmental pollution related to energy consumption. As shown in Figure 5b, *EC* has decreasing contribution rate to unit standard deviation in *NU* for provinces in the central, northeastern, and western regions, yet *EC* has an increasing contribution rate to unit standard deviation in *NU* for provinces in the eastern region, which became the largest after the tenth forecast period. These results also imply that environmental pollution related to energy consumption may gradually increase as the pace of new-type urbanization accelerates for provinces in the central, northeastern, and western regions, which are consistent with the estimation results of the PVAR model.

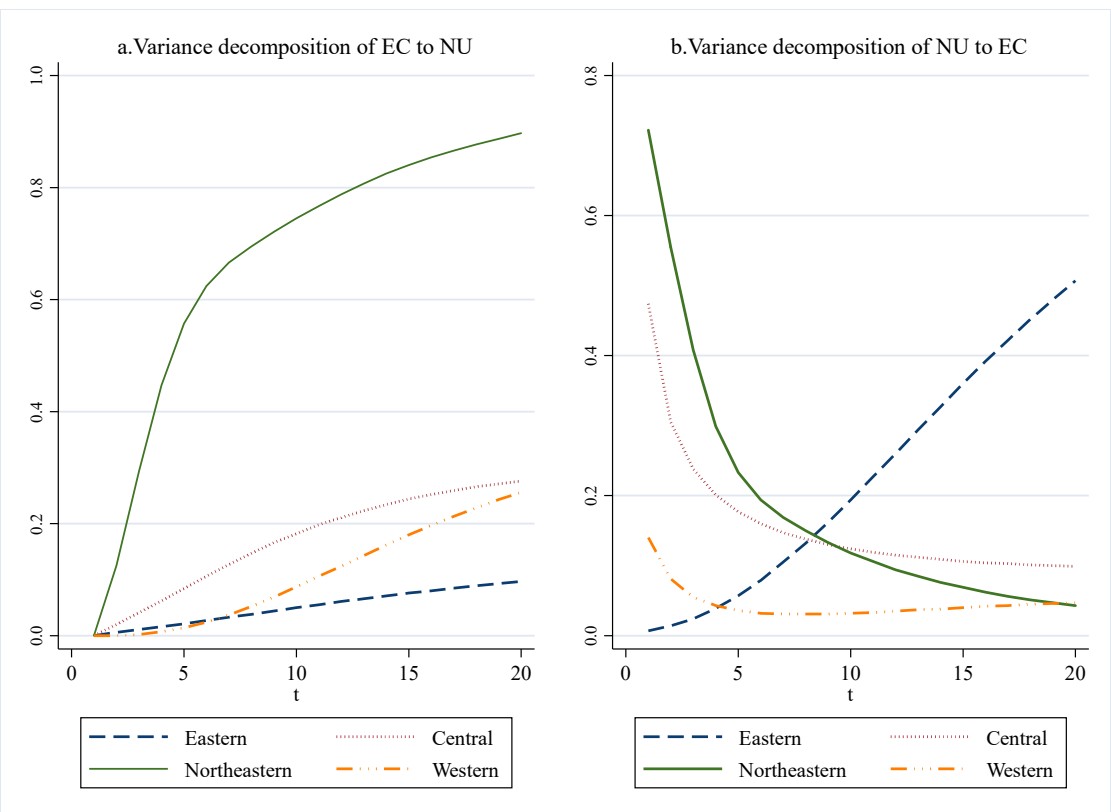

**Figure 5.** Variance decomposition of *EC* and *NU*.

## 4. In-Depth Analysis of Energy-Related Data

Finally, the pollution-related calculations are included for an in-depth analysis. We collect supplementary data, such as carbon emissions per real CNY 10,000, energy used per real CNY 10,000 (energy consumption intensity), the ratio of output value in the tertiary industry to real GDP, and the average per capita real GDP. All the relevant data are from each year's China Statistical Yearbooks, China Energy Statistical Yearbooks, each provincial statistical yearbook, and so on. As original data for the carbon emissions can be obtained from 2003 to 2019, the sample data span from 2003 to 2019. Figure 6 shows these calculations for different groups: carbon emissions (Figure 6a), energy consumption intensity (Figure 6b), industrial structure (Figure 6c), and per capita real GDP (Figure 6d), respectively.

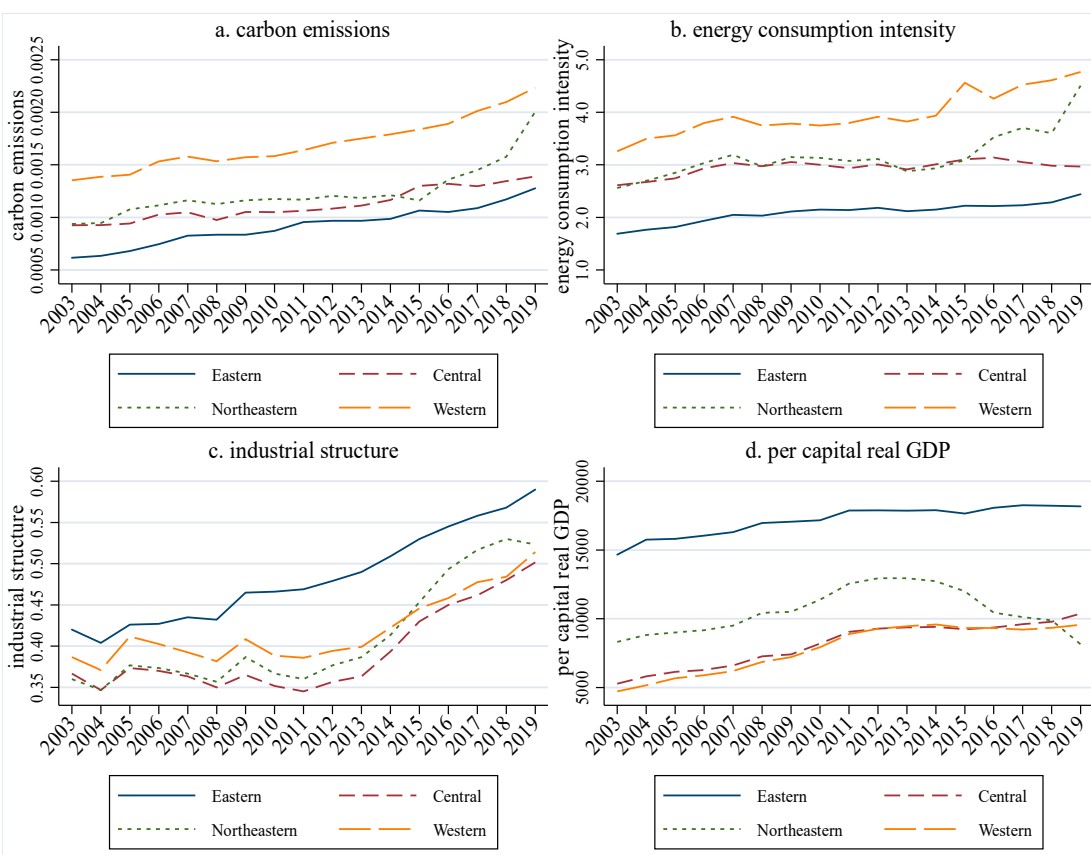

**Figure 6.** Some energy-related data.

As shown in Figure 6a, the largest carbon emitter appears in the western region, followed by the northeastern and central regions, and the least carbon emitter happens to be in the eastern region. As the pace of the new-type urbanization accelerates, provinces in the eastern region may make great strides to reduce the pollution of carbon emissions related to energy consumption. These results undoubtedly denote that there is an EKC relation. As shown in Figure 6b, the western region has the highest energy consumption intensity, followed by the northeastern and central regions, and finally the eastern region. As energy consumption intensity represents the efficiency of energy use, the characteristics of this variable in the four different groups denote that the energy is the most efficiently used in the eastern regions, followed by central and northeastern regions, and finally the western region. As the pace of new-type urbanization accelerates, provinces in the eastern region may successfully change their energy mix, and a decrease in carbon emissions is also can be anticipated. For example, pollution-free electrical energy can be massively put to use. To sum up, these two variables both confirm the causal relationship that new-type urbanization leads to energy consumption negatively for provinces in the eastern region, and new-type urbanization leads to energy consumption positively for provinces in the central, northeastern, and western regions over time, which seems to be in agreement with the estimation results of the PVAR model.

Figure 6c,d further provides the economic reality for the EKC relation and the causal relationship between the two. As shown in Figure 6c, the ratio of output value in the tertiary industry to real GDP in the eastern region is the highest among the four groups, followed by the northeastern region, the western region, and the central region. As shown in Figure 6d, per capita real GDP in the four groups is the same as Figure 6c. As the economy develops in the eastern region, there may be a trend toward the tertiary industry, which produces low-pollution products. Therefore, new-type urbanization leads to negative energy consumption. Provinces in the central, northeastern, and western regions are

generally eager to raise the per capita real GDP level, so standards are lax concerning relevant environmental regulations. Even some high-pollution industries are encouraged in these regions, and spontaneously their ratio of output value in the tertiary industry to real GDP is behind the eastern region. Hence, the economic reality is consistent with the EKC relation and the causal relationship between the two.

## 5. Concluding Remarks

The panel data of Chinese provincial administrative units are classified into four categories according to the widely adopted standard in this paper. In order to explore the causal relationship between new-type urbanization and energy consumption, we employ the PVAR model to investigate this issue and obtain the following conclusions. (1) For provinces in the eastern region, new-type urbanization leads to energy consumption negatively, and energy consumption leads to new-type urbanization positively, which becomes negative over time. For the provinces in the central and northeastern regions, an increase in the level of new-type urbanization does not bring about energy consumption, while the uni-directional causal relationship running from energy consumption to new-type urbanization is similar to provinces in the eastern region. For provinces in the western region, there is negative feedback causality between the two. (2) For provinces in the eastern region, the advantage of energy consumption is larger than its disadvantage in the short run, yet the relationship is the opposite in the long run. For provinces in the central and northeastern regions, the advantages of energy consumption and its disadvantages are similar to provinces in the eastern region. However, for provinces in the western region, the disadvantage of energy consumption may exceed its advantage. These findings are also consistent with the EKC relation. Once a province reaches a high level of new-type urbanization, it may conversely reduce the negative externality of environmental damage related to energy consumption. (3) The impulse response analysis further presents the dynamic relationship between the two. The variance decomposition demonstrates that the effect of energy consumption on new-type is much larger than the effect of new-type urbanization on energy consumption for the provinces in the eastern region, while it is totally the opposite for provinces in the central, northeastern, and western regions. (4) The largest carbon emitter appears in the western region, followed by the northeastern and central regions, and the smallest carbon emitter happens to be in the eastern region. The western region has the highest energy consumption intensity, followed by the northeastern and central regions, and finally the eastern region. The ratio of output value in tertiary industry to real GDP in the eastern region is the highest among the four groups, followed by the northeastern region, the western region, and the central region, and the per capita real GDP in the four groups is the same as the ratio of output value in the tertiary industry to real GDP.

On the basis of our conclusions above, several straightforward policy implications can be put forward. As the causal relationship presents regional heterogeneity, a one-size-for-all energy policy will not work effectively. Our government should take into consideration the different levels of new-type urbanization while implementing an energy consumption policy. Specifically, for provinces in the eastern region, they should spare no effort to promote the level of new-type urbanization and mitigate energy use in the construction of new-type urbanization. For provinces in the central region, energy consumption is not conducive to promoting the level of new-type urbanization, hence they should adopt some conservation policies to avoid the negative externality of environmental damage related to energy consumption. For provinces in the northeastern and western regions, the negative externality of environmental damage related to energy consumption may gradually increase, and they should especially pay attention to cope with the possible environmental damage and take remedial actions. Finally, according to the in-depth analysis of energy-related data, for the provinces in the northeastern and western regions, they should try to raise the efficiency of energy use, so as to reduce carbon emissions. For example, our Chinese government can encourage enterprises to carry out energy-

saving technologies in the production of goods and services. Additionally, for provinces in the northeastern and western regions, some high-pollution industries may have been encouraged, hence standards have to be strict concerning environmental protection and relevant environmental regulations.

The used methodology in this paper has some shortcomings, which calls for further research. Firstly, the constructed comprehensive urbanization index does not take into consideration the aspect of income inequality due to data availability. A comprehensive urbanization index should be constructed to reflect more about social welfare. Secondly, the empirical study in this paper does not consider the spatial spillover effect when exploring the causal relationship between energy consumption and urbanization. As we all know, both the level of energy consumption and urbanization probably present a spatial autocorrelation relationship, so future studies should employ spatial econometric techniques in this topic to deal with the possible estimation bias of the non-spatial econometric models. Thirdly, the empirical study in this paper is still based on Chinese provincial-level data. As the urban areas are the primary body to promote the urbanization process currently, it is of great value to conduct in-depth investigations at the city level.

**Author Contributions:** Conceptualization, Y.Q.; methodology, Y.Q.; formal analysis, C.C. and Y.G.; data curation, Y.G.; writing—original draft preparation, C.C.; writing—review and editing, C.C. All authors have read and agreed to the published version of the manuscript.

**Funding:** This research received no external funding.

**Institutional Review Board Statement:** Not applicable.

**Informed Consent Statement:** Not applicable.

**Data Availability Statement:** The data presented in this study are available on request from the corresponding author.

**Conflicts of Interest:** The authors declare no conflict of interest.

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
