# Peer review of "A Causal Relationship between the New-Type Urbanization and Energy Consumption in China: A Panel VAR Approach"

_sustainability, doi:10.3390/su151411117_

Round 1

Reviewer 1 Report

The paper attempts to explore the causal relationship between China's new urbanization and energy consumption. Based on panel data, regression analysis is conducted. The paper has the following problems:

1) author's purpose was to explore causal relationships, but only regression statistics were conducted on the correlation between urbanization data and energy consumption data. Due to the influence of many factors such as energy structure, energy technology, and industrial structure on energy consumption, some of which even exceeded the impact of urbanization, the author was unable to reveal a more profound causal relationship;

2) only conducting correlation analysis between urbanization data and energy consumption data is a very conventional approach that does not bring new discoveries to readers. Even though the author explains that more interpretations have been made, these interpretations are still evident, and the paper does not propose significant technological innovation or new explanatory perspectives.

Reviewer 2 Report

The manuscript “Causal relationship between the new-type urbanization and energy consumption in China: A panel VAR approach” explored the relationship between the urbanization process and energy consumption in China during two decades (1999-2020) using pre-defined indices and the PVAR model. Some minor remarks are:

- expand keywords, for example, add “PVAR approach”,

- the urbanization index (the level of new-type urbanization indicator) should be explained in more detail, even authors stated that want to “save space”, it would be very useful to understand in detail what exactly that index is.

- It will be interesting to combine Figures 1 and 2 in one. For example, the color method is used for energy consumption and different lines for urbanization (to overlap two phenomena).

- According to Figure 2, the entire territory of China has the highest level of the new-type urbanization index in 2020. I would ask the authors to discuss it. Are there any methodological shortcomings reflected there?

- Figure 3 clearly demonstrates that the correlation coefficient is not an appropriate choice for exploring these variables. Consider removing this part.

- In most graphs, the labels of the axis are missing.

- All footnotes do not seem to be necessary. Replace that part with the main text.

- It is necessary to point out the shortcomings of the used methodology, which are limitations and shortcomings.

Reviewer 3 Report

Well done. Well written and suitable approach for the goal of the research. It is fine for publication.

Reviewer 4 Report

The article examines an empirical study of the causal relationship between the urbanization of a new type and energy consumption using the administrative units of the provinces of China. The strength of this article is a well-researched literature review, as well as empirical research. The work is of interest and deserves further research and expansion. Despite this, some provisions were found in the work that require additional explanations. This is described below.

 The first section of the article is traditionally an introduction. Here, the authors reveal the relevance of the research topic, identify the problems. The authors note that China's urbanization significantly improves people's living standards, but is also considered the main reason for the increasingly serious increase in energy consumption and, of course, leads to environmental pollution. Therefore, two issues are in the balance: urbanization and energy consumption. The authors analyze the early works in sufficient detail, which allows us to judge the depth of the study. The section of the introduction ends with the designation of the authors' contribution to science, the scientific novelty is clearly expressed. Therefore, the introduction is a well-developed section of this article. In the second section, the authors logically cite the methods of empirical measurements on energy consumption, the urbanization index, as well as the results of measurements. The sample data from the empirical study covers 22 years from 1999 to 2020. Given the rapid growth of urbanization, especially in China, such coverage is quite presentable. This can be judged from the results obtained in Figures 1 and 2. In Figure 3, the authors demonstrate the results of processing statistical data on the growth of energy consumption and the growth of the index of integrated urbanization. According to the dependencies, especially in Figure 3d, the correlation coefficients show a weak relationship. Here, the authors need to explain in more detail what the established weak correlation may be connected with? What factors influenced the result? Also, the authors do not describe how many parallel experiments were made? What percentage of error was obtained during data processing?

The third section, the authors present the results and analysis of the regression of the panel vector autoregression (PVAR) model. To conduct an empirical study, a PVAR model is used. The authors check the stationarity of panel data and the causal relationship between the level of urbanization of the new type and energy consumption, choose the optimal delay order, build an impulse response graph based on the regression of the PVAR model, and perform dispersion decomposition. According to the results, it can be found that some dependencies are unstable, the graph has function optimums, sharp jumps. How can we explain this kind of change in the dependencies in the impulse response figures (Figure 4) and dispersion decomposition (Figure 5)? In the fourth section, the authors pay special attention to the issue of environmental pollution within the framework of this study. The results are presented as dependencies. However, as in the previous dependencies, function optimums are observed. I would like to know the authors' comments on this. In the fifth section, the authors provide concluding observations and conclusions within the framework of the study. The authors obtained interesting results that suggest that the impact of urbanization on energy consumption will be different in different regions. At the end of the section, the authors provide research perspectives. In my opinion, the work needs to be continued research, where it is necessary to consider the impact of individual production sectors or groups, factories, and so on, to assess their relationship with urbanization and energy consumption: Badriev A.I, Sharifullin V.N. About manufacture of electric energy at the chemical enterprises. Journal of Computational and Theoretical Nanoscience, 2019, vol.16, Is.1, pp.209-212.

Round 2

Reviewer 4 Report

I recommend that you correct the formatting of the reference list and expand it.

Author Response

We have corrected the formatting of the reference list.